Diffusion tubes: a method for the mass culture of ctenophores and other pelagic marine invertebrates

Patry Wyatt L. wpatry@mbayaq.org
Bubel MacKenzie
Hansen Cypress
Knowles Thomas
Animal Care Division, Monterey Bay Aquarium , Monterey, CA , USA
Esteban María Ángeles
Electronic publication date: 2020 Apr 7
Publication date: 2020
Volume: 8
Electronic Location ID: e8938
Received 2019 Oct 11; Accepted 2020 Mar 18
Copyright: © 2020 Patry et al.
Copyright year: 2020
Copyright holder: Patry et al.
License: This is an open access article distributed under the terms of the Creative Commons Attribution License, which permits unrestricted use, distribution, reproduction and adaptation in any medium and for any purpose provided that it is properly attributed. For attribution, the original author(s), title, publication source (PeerJ) and either DOI or URL of the article must be cited.
License URL: https://creativecommons.org/licenses/by/4.0/

Keywords: Ctenophore culture, Siphonophore culture, Model organism system, Larvacean culture, Aquaculture, Diffusion tubes, Hormiphora, Bolinopsis, Leucothea, Pleurobrachia

Funding: Monterey Bay Aquarium Foundation Funding was provided solely by the Monterey Bay Aquarium Foundation. The funders had no role in study design, data collection and analysis, decision to publish, or preparation of the manuscript.

==============================
The culture of pelagic marine invertebrates, especially the ctenophore Mnemiopsis leidyi, has been demonstrated in past studies dating back to the 1960s; however, the mass culture of delicate pelagic invertebrates has remained elusive. By using a pair of acrylic tubes and enabling water diffusion between them, we have been able to reliably and cost effectively mass culture several genera of ctenophores (Pleurobrachia, Hormiphora, Bolinopsis, Mnemiopsis and Leucothea), one species of siphonophore (Nanomia) and one species of larvacean (Oikopleura). The simple, compact method is effective enough to support two permanent exhibits of ctenophores at the Monterey Bay Aquarium while minimizing live food culture requirements with the potential to support further investigation of pelagic marine invertebrate ontogeny, ecology and genomics.

Introduction

Interest in ctenophore culture and other pelagic marine invertebrates as model organisms has surged recently (Howes et al., 2014; Jaspers, Møller & Kiørboe, 2015; Jaspers, Marty & Kiørboe, 2018; Martí-Solans et al., 2015; Presnell & Browne, 2019). Basic ontogeny of ctenophores has been observed since at least the turn of the 19th century by Chun, Agassiz and Mayer however maintaining cultures in the laboratory remained elusive until later in the 20th century. Pioneering efforts in developing rearing vessels for pelagic culture were made by Wulf Greve with his invention of the planktonkreisel (Greve, 1968, 1970) and other variations such as the double cuvette design (Greve, 1975). Greve achieved several generations of Pleurobrachia pileus in these studies. The kreisel was later formalized by Hamner (1990) for use aboard ships and then adapted for use in public aquariums by the Monterey Bay Aquarium (Raskoff et al., 2003).

The method presented here originated at the Monterey Bay Aquarium in July 2015 to support the temporary exhibition, The Jellies Experience and while collaborating with W. E. Browne of University of Miami on M. leidyi culture (Presnell et al., in press). After successfully culturing Mnemiopsis, we turned our focus to culturing other ctenophores such as Pleurobrachia bachei and Bolinopsis infundibulum. The young cydippids of these species are so delicate that they do not survive in traditional plankton kreisels. They are susceptible to being damaged by the seawater flow in the tank and sticking to the outflow screen. Additionally, juveniles do not survive in standing dishes of seawater due to rapid accumulation of ammonia and high mortality from being transferred into new seawater. Therefore, we needed a rearing tank with minimal or no mechanical driven water flow, no outward pressure on the outflow screen and the ability to passively exchange new seawater. We experimented with rearing tanks featuring seawater flow just outside of the outflow screen, such that seawater could passively exchange without creating outward pressure on the screen. While these methods were effective at eliminating buildup of nitrogenous wastes and minimizing forces on juvenile ctenophores caused by seawater flow, the tank shapes were not appropriate for the swimming and feeding behavior of the animals. In traditional rectangular aquaria, juveniles were observed actively consuming prey at or near the surface and then sinking to the bottom, where significant biofouling occurs. Initially we designed and constructed a cylindrical tank with more vertical space for feeding, minimal biofouling surface area with mesh covering the bottom and secured in a pseudo-kreisel such that water flowed up through the bottom. This initial design proved somewhat effective, however juveniles still spent significant time contacting the bottom mesh. An improved design providing even more vertical space, while reducing the volume of the tank was implemented. Our final version utilizes a double tube configuration in which an elongated inner tube is surrounded by an outer tube “sleeve”, affectionately referred to as a “cteno-tube” in the lab.

Over the past 4 years, we have experimented with this new double tube rearing tank design (Fig. 1) and have refined a protocol for raising fragile pelagic ctenophores. We also performed a single experiment using several flow rate treatments to observe a number of physical parameters (pH, oxygen and temperature) within the “cteno-tube” in relation to adult ctenophore yield. Here we present these results, additional variations on the double-tube method and provide a best practices protocol for the mass culture of pelagic ctenophores and other gelata.

Figure 1 Experimental setup using three diffusion tubes and 3D rendering of diffusion tubes.

(A) Three sets of diffusion tubes on a wet table, (B) 3D rendering of both tubes with supply tubing at top right and (C) outer tube has been removed to reveal the inner tube resting on a riser made with rigid plastic mesh.

Method

Materials

Diffusion tubes were constructed using two 0.9 m long acrylic tubes. The outer and inner tubes had volumes of ~35 L and 28 L with inner diameters of 21.59 cm and 20 cm respectively, so that one could be placed inside the other (an inner and outer tube, Figs. 2A and 2B). Each acrylic tube had a thickness of 0.3175 cm. The bottom of the inner tube was fitted with 55 µm nylon screening (Model# M55; PentairAES, Inc., Cary, NC, USA, Fig. 2E) using silicone sealant (Dowsil 795 or 999-A, Fig. 2J). This inner tube was then placed on a 3 cm tall riser made of rigid plastic mesh (Model# N1020; PentairAES, Inc., Cary, NC, USA, Fig. 2G) such that the top of the inner tube rises above the outer tube (Fig. 2, Step 7). The outer tube was glued to a ~1 cm thick square acrylic baseplate (Fig. 2C), 15–20 cm2, using acrylic cement (Weld-On #16 Fast Set solvent cement, Fig. 2D). Rigid PVC tubing, 0.635 cm inner diameter (Part# 48855K41; McMaster-Carr, Elmhurst, IL, USA) was placed in the outer tube, between tubes, providing water flow near the bottom. Filtered seawater (5 µm) was pumped through the tubing at rates of 1.1, 2.2 and 4.5 liters per minute (Lpm). Three identical pairs of diffusion tubes were constructed and placed on a wet table with recirculating filtered seawater (5 µm) such that water overflowed out of the outer cylinder and on to the table. Diffusion between tubes through the 55 µm nylon screening was observed by the addition of dye (McCormick blue food coloring) to the incoming water. In the 4.5 Lpm treatment, flow was such that a slight suction vortex pulled the ctenophore eggs into the bottom screen which prevented hatching. To resolve this, a PVC pipe tee fitting (Part# 4881K47; McMaster-Carr, Elmhurst, IL, USA) was added to the bottom of the inflow tube to reduce water velocity and divert the incoming water away from directly underneath the screen, which proved successful. All current materials and pricing are listed on the “CtenoTube” repository Wiki on GitHub (https://github.com/wyattp11/CtenoTube/wiki) as of this writing the cost for a single diffusion tube setup is ~$465 including all tools necessary for assembly.

Figure 2 Assembly instructions for diffusion tubes.

(A) Materials needed for construction. (B) Construction of the outer tube. (C) How to cut the screen for the inner tube. (D) Prepare inner tube with silicone sealant. (E) Secure the screen to the inner tube using the rubber band. (F) Construction of the riser. (G) Place riser inside outer tube. (H) Place inner tube inside outer tube.

Assembly instructions

Outer tube construction (Fig. 2B): center the tube with the widest internal diameter (a) on the base plate (c). Use the syringe (i) to draw up some acrylic glue into the syringe. Inject acrylic glue into the seam between (a) and (c). Leave the tube and base plate to cure for at least 24 h before handling again. Cut screen (Fig. 2C): cut a circular piece from the sheet of screening (e) using scissors (h). The diameter should be at least 2 cm + the diameter of the inner tube (b) to allow enough overlap to glue down. Glue inner tube (Fig. 2D): Place a bead of silicone sealant (j) around the top edge of the inner tube (b) and then about 1 cm below the edge (red lines). Smear the lower bead so that it is smooth and flat using a gloved finger, ice or other flat edge. Secure mesh on inner tube (Fig. 2E): Use the rubber band (f) to secure the piece of cut screen from Step 2 over the end of inner tube with the silicone sealant (j). Place another bead of silicone over the mesh and smooth until the screening is no longer visible. The edge of the tube can also receive a light bead and smoothing. Construct riser (Fig. 2F): Cut a 3 cm tall length of the rigid plastic mesh (g) that matches the circumference of the inner tube (b). Remove any sharp edges, edges should be smooth. Use a pair of small zip ties (k) to secure the riser to itself, forming a tube. To avoid the riser puncturing the fine screen (e) a piece of cut rubber tubing may be placed on the top edge of the riser. Place the finished riser at the bottom of the outer tube (Fig. 2G). Now place the inner tube (b) inside the outer tube (a) so it gently rests on the riser (Fig. 2F). The tube setup is now ready for filtered seawater. The supply line may be rigid acrylic tubing or flexible PVC tubing.

Characterizing physical parameters

A set of probes (Hach HQ40d with LDO probe) for measuring pH, dissolved oxygen and temperature was deployed within the inner cylinder of each set of tubes and reads were taken every 12.7 cm at seven discrete depths. Tubes were allowed to equilibrate for at least 24 h prior to testing and the probes were lowered slowly to read depths using a simple pulley system. A minimum settling time of 2 min at each depth was necessary to take accurate reads of each parameter. We also used a dye to observe mixing/stratification within the tube and found that the inflow tube caused flow disturbance resulting in water primarily diffusing in from the opposite side of the tube inflow. In order to account for discrepancies caused by the inflow tube’s position on one side vs. the other, probe readings were established twice, once closest to the inflow tube and once on the opposite side.

Methods of spawning and culturing

Three adult Hormiphora californensis were placed in each inner cylinder (Fig. 3A). To induce spawning (Fig. 3B), the tubes were shrouded in a black, opaque cover for total darkness overnight, followed by 2 h of bright light the next morning (Kessil A360 Tuna Blue LED aquarium light placed ~5 cm above the inner cylinder) (Pang & Martindale, 2008). When eggs were observed throughout the water column, usually ~12 h post spawn or 24 h after setup, the adults were removed from the tubes using a spoon attached to a short pole or by slowly swirling them to the surface.

Figure 3 Spawning ctenophores in the diffusion tubes.

(A) Hormiphora californensis spawning in the diffusion tube with a bright LED light. (B) Pleurobrachia bachei releasing sperm from the meridional canals beneath the ctene rows.

Hatching of the eggs was observed 12–48 h post spawning. Juvenile cydippids were fed on the first day post hatch (dph). Each tube was given a standardized concentration (~30 nauplii mL−1) of live Parvocalanus crassirostris (Reed Mariculture Inc., Campbell, CA, USA) nauplii mixed with live algae (Isochrysis galbana, Rhodomonas lens, Dunaliella tertiolecta, or Tetraselmis chuii, Florida Aqua Farms Inc., Dade City, FL, USA) in a 1:1 ratio, both cultured in the lab. P. crassirostris nauplii were added at the surface of the tube water column for 6 weeks using the following regimen: 25 mL three times per week (Sunday, Tuesday, Thursday) for the first 14 days, increasing to 50 mL for the next 14 days and 100 mL for the last 14 days.

The ctenophores were observed throughout the water column and sampled once a week for 6 weeks to be photographed and measured. For photo documentation and measurement, 15–30 juvenile cydippids were randomly pipetted out of the water column down to 30 cm depth and transferred to a glass crystallizing dish filled with filtered seawater for photographing under a microscope (Zeiss Stereo Zoom V16, Canon EOS Rebel T5i). Each photograph was scaled and measured using ImageJ 1.52A (Schneider, Rasband & Eliceiri, 2012).

Post-diffusion tube rearing

When cydippids reached ~0.5–1 cm in diameter (~30 dph) they were transferred using widened disposable polyethylene pipettes (tips cut off, Transfer pipets 13-711-7M; FisherBrand, Pittsburgh, PA, USA) or small polypropylene beakers (50–100 mL Nalgene Griffin low form beaker) from the diffusion tubes to kreisel tanks (Raskoff et al., 2003) modified to provide for better longevity of ctenophores. The kreisel screens were replaced with a polyethylene mesh with 4 mm openings (Model #N1670; PentairAES, Inc., Cary, NC, USA). Slot openings in the supply boxes were reduced in width using a single piece of 4 mm corrugated plastic (high-density polyethylene and polypropylene, Model #24244SC; CorrugatedPlastics.Net, Dongguan, Guangdong, China) instead of two 6 mm pieces which are typically found as filler for 12.7 mm slot openings. Pumps were set to provide ~10 Lpm of volumetric flow to the lower supply box and ~7.5 Lpm to the upper box (Model DCW-2000; Jebao, Inc., Camrose, AB, Canada) using flow meters (Model #F-44500L-8; Blue-White Industries, Ltd., Huntington Beach, CA, USA). The upper box provided ~0.8 Lpm of filtered seawater (5 µm) at 13 °C. The diet was diversified to include 1–2 mysid shrimp (Mysidopsis bahia, Aquatic Indicators Inc., St. Augustine, FL, USA) per comb jelly, usually ~200 per kreisel every 2 days and 200 mL of P. crassirostris adult copepods fed daily around noon.

Non-ctenophore rearing

Oikopleura sp. and Nanomia bijuga were reared in much the same manner as the ctenophores. Up to 10 adults of Oikopleura were used to start a culture in a diffusion tube setup, however this is not typically feasible with Nanomia due to the length of the wild colonies which will tangle when more than 3–4 adults are added to an inner tube of 15 cm diameter. If adult Nanomia are to be retained after spawning, then the number of adults should be limited based on the diameter of inner tube used. Culture temperature for Nanomia was manipulated, adults were spawned at 13 °C and then the temperature raised to 18 °C for the planulae to begin development. At colder temperatures Nanomia had very low colony development (14 °C) or no development whatsoever (7 °C). Once multi-tentacle Nanomia reach ~5 cm non-contracted length they can be acclimated back down to 14 °C and graduated to a pseudo-kreisel (traditional kreisels do not seem to work well for adult Nanomia as the pneumatophores collide with curved acrylic surfaces too often, while they are well adapted to touching the surface of the water in a pseudo-kreisel). Adult Nanomia required live mysids (Mysidopsis bahia) while in the pseudo-kreisel to reach maturity, which occurs rapidly over the course of a week at 14 °C. As of this writing, F1 Nanomia (ranging in non-contracted lengths of 15–25 cm) were spawned and successfully to F2 individuals using the proposed protocol (see “Conclusions”). Oikopleura culture was performed just once at 13 °C and a new generation appeared every week with regular feedings of live algae however it was unclear if 3 or 4 generations were produced due to the rapid succession. It is not necessary to remove adult Oikopleura after spawning however it may be advantageous to remove discarded mucous houses. At the end of ~30 days the screen had clogged with algae, perhaps mucous and the culture was lost. Neither Oikopleura or Nanomia were part of the experimental results presented in this article.

Statistical analysis

Mean length data was analyzed using R (R Core Team, 2017). Growth rates were adapted to the data using the R package Growthcurver (Sprouffske & Wagner, 2016, https://cran.r-project.org/package=growthcurver). Descriptive statistics, Kruskal–Wallis ANOVA and Wilcoxon pairwise comparisons were made using packages dplyr (Wickham et al., 2018), ggpubr (Kassambara, 2019) and ggplot2 (Wickham, 2016).

Results

Physical parameters and depth profiling

Temperature

Under the highest turnover rate (4.5 Lpm), the water column exhibited a significant thermocline within 12–24 cm of the surface. With a surface temperature between 16 and 17 °C, the surface layer was approximately 3–4 degrees warmer than the water below 24 cm in depth (Fig. 4A). This was in agreement with dye observations, which indicated slower turnover in the top half of the water column under the 4.5 Lpm treatment. Although the other two treatments exhibited less dramatic thermoclines, the mean surface temperature exceeded the mean bottom temperature by at least 1.5 °C in all three treatments. Temperatures reported are the mean of the measurements taken on the two opposite sides of the inner cylinder, nearest and farthest from the inflow line. No significant difference was found between the near and far side measurements.

Figure 4 Depth profiles of each treatment for temperature, pH and dissolved oxygen.

Line graph showing the means (points) of the two measurements taken on opposing sides of the inner tube. (A–C) Temperature by treatment. (D–F) pH by treatment. (G–I) Dissolved oxygen by treatment.

pH

pH was the most consistent of all parameters measured, ranging between 7.84 and 7.90 across all flow rates and depths, with the low flow treatment yielding the lowest pH at the surface of the water column (Fig. 4B). The low and high flow treatments exhibited slightly decreased pH at the surface layer.

Dissolved oxygen

Dissolved oxygen measurements exhibited no significant differences with regard to depth, flow rate, or temperature. The high flow treatment showed slightly decreased dissolved oxygen at the surface layer in conjunction with decreased pH, however the measurements were within the standard error (±0.02 pH, ±0.2 mg/L DO) of the probe (Fig. 4C).

Culturing success

All three treatments were successful in raising >50 H. californensis over a 6-week period. The high flow (4.5 Lpm) treatment yielded the most cydippids (89), followed by medium flow (2.2 Lpm) (87) and low flow (1.1 Lpm) (75). The mean cydippid length of each treatment varied little from 0–23 dph (each p > 0.07) and then greatly increased by 30 dph (Fig. 5). Cydippids in the high flow treatment had the highest mean lengths at the end of the experiment (Wilcoxon signed-rank test, p = 0.0046 and 2.1 E−4, low vs. high and medium vs. high respectively), while the medium treatment had the smallest mean lengths (Fig. 6). Growthcurver produced best fits and growth rates for the low and high flow treatments but failed to find a best fit for the medium flow treatment (Table 1).

Figure 5 Total length measured over 35 days.

Mean lengths (points) of H. californensis with SE (whiskers) in low, medium and high flow treatments.

Figure 6 Pairwise comparison of means for low, medium and high flow treatments at 35 dph.

At 35 dph the high and low flow treatments ended with significantly greater mean lengths (points) than the medium flow treatment. The high flow treatment had the greatest mean lengths. Whiskers represent 95% CI.

Table 1 Growth rates produced using R package: growthcurver.

Treatment	K	N	r	
Low (1.1 Lpm)	33.759	0.013	0.635	
Med (2.2 Lpm)	–	–	–	
High (4.5 Lpm)	33.705	0.026	0.574	

Discussion

Our method utilizes a cylindrical tank shape which maximizes the ratio of water volume to the surface area of tank bottom, which has several important effects. By minimizing the bottom biofouling area, contact with these fouled surfaces is greatly decreased, vertical swimming or migration space for juveniles is maximized and a thermocline is produced where the juvenile cydippids can feed furthest from the tank bottom. Another key feature of our design is the passive exchange of seawater through the mesh which generates much lower water velocities than in a traditional plankton kreisel. These lower flow rates allow delicate juvenile ctenophores to develop in a very gentle environment while still facilitating the exchange of clean filtered seawater. In contrast to traditional aquaria for gelatinous zooplankton, our method provides a superior pelagic environment for young ctenophores and other gelata larvae that is, inexpensive and easily replicable. At the time of this writing a single diffusion tube setup costs ~$465 USD, a small pseudo-kreisel (<0.5 m D) can cost twice that of a diffusion tube setup and takes up 2–3 times as much surface area in the lab which makes replication challenging. Reliable, replicable mass culture of ctenophores has also not been demonstrated in traditional kreisels yet.

Using the highest turnover rate generated a more pronounced vertical thermocline and resulted in higher survival and significantly faster growth rates among the treatments. It is not clear why the low flow outperformed the medium flow treatment in mean length at 35 dph. Flow rate was a critical factor in our previous runs with the diffusion tubes, if the flow rate was too low, we observed pre-mature die-off of the culture. So, it may be that the medium and low flows are actually equal in performance and the higher flow rate the ideal. Replication of this experiment should produce a clear quantitative answer to this specific problem. While the method has been replicated many times, this specific experiment testing flow was only performed once.

Overall, fewer H. californensis were produced in this experiment than previous and subsequent trials with the diffusion tubes. An F4 population was achieved shortly after the experiment using 10 parents of the F3 population, yielding >600 cydippids and a similar run with 10 F4 parents produced >500 cydippids to create the F5 generation currently in culture (growth and flow data were not collected for these runs). We believe this was primarily due to the restricted quantities of food used for this experiment (in previous use of this method feeding amounts varied between just 25 mL up to 200 mL of copepod nauplii, depending on the specific user). When feeding quantity was increased from 50 mL to 100 mL at week 5 there was a notable increase in growth rate across treatments at 40 dph. It is also possible that lower yields may reflect use of fewer parents (n = 3) than previous/subsequent trials (n = 10) due to inbreeding depression from more self-fertilization. Ultimately, we have adopted a feeding regime that doubled the amounts of copepod nauplii and uses 10 adults to keep the yield higher (see “Conclusions”).

Further development of this method has the potential for increased efficiency and work with new species of gelata. Changing cylinder height and width may allow for more variation in thermocline profiles and vertical space, however bigger tubes will require more volumetric flow to the outer cylinder. For larger ctenophore species diffusion tubes with larger diameters may have increased utility (Table 2), for example, Leucothea pulchra was successfully cultured to F2 using diffusion tubes 2 m high and 0.35 m diameter with a flow of ~4 Lpm (Fig. 7). Bolinopsis infundibulum and Nanomia bijuga were grown using an inner cylinder measuring 0.9 m high and 0.25 m diameter (outer cylinder was 0.9 m high and 0.30 m diameter) (Fig. 8). Changing the mesh size may impact diffusion rates between the two tubes, thermocline formation and perhaps be better fitted to a particular target species. For example, a 22 µm mesh was originally used for P. bachei but we found the mesh fouled quickly. When replaced with 55 µm mesh, diffusion and flow performance were enhanced and fouling no longer a problem.

Table 2 Suggested diffusion tube sizes (mm) and flow (Lpm) by species.

Species	Inner tube	Thickness	Outer tube	Thickness	Height	Flow	
Hormiphora californensis	203	3.175	222	3.175	900	5	
Pleurobrachia bachei	127	3.175	145	3.175	900	5	
Bolinopsis infundibulum	305	3.175	254	3.175	600	7	
Leucothea pulchra	350	3.175	381	6.35	2,000	10	

Figure 7 Taller diffusion tube setup for Leucothea pulchra.

(A) Two meter tall diffusion tubes for rearing Leucothea pulchra sitting inside of a 200 L reservoir, (B) three large adults were spawned to produce the F1 generation, (C) thermocline visible at the top of tube, 9 dph and (D) various stages of F1 larval L. pulchra visible at the surface of the inner tube, 21 dph.

Figure 8 Shorter and wider diffusion tubes for other species of gelata.

(A) 30 cm wide diffusion tubes used for Bolinopsis infundibulum and Nanomia bijuga culture, (B) larval N. bijuga at 35 days post settlement visible near the top of the tube and (C) mature B. infundibulum spawning under bright light, the wider and shorter tube makes it easier to place and remove the adult lobates.

Conclusions

After 4 years of trials culturing various gelata using different sizes of the diffusion tubes, we derived a standard procedure for mass culture of ctenophores from 1 to 10 parental adults using 1 m diffusion tubes (Hormiphora and Pleurobrachia) and a feeding regime based on results yielding >100 adult ctenophores in about 30 days was adopted as follows: To start a culture from wild collected adults, use 1–10 adult ctenophores depending on desired genetic diversity. Adults should be fed a fish or mysid meal 24 h prior to spawning and again 1 h before the dark period begins (Presnell et al., in press).

To induce spawning, place all adults in an equilibrated diffusion cylinder (for inner diameter 20 cm, a flow rate of 4.5 Lpm), shrouded in opaque material to ensure complete darkness for a 4 h period (or overnight). Uncover the tube and expose to bright lighting for at least 2 h (Presnell et al., in press).

When adults have spawned and eggs are visible in the water column, gently remove the adults from the tube using ladles or dishes.

Eggs hatch within 12–48 h, first feeding should ideally take place within the first 24 h of hatching to ensure that cydippids are able to begin feeding immediately.

Feeding schedule: see Fig. 9.

At 30 dph the cydippids may be moved to a pseudo-kreisel or kreisel tank. If cydippids have not achieved a total body size of 0.5–1 cm diameter by this time, their residence in the tube may be increased to as long as 40 dph. However uniformly slow growth rates are a sign of a weak hatch, not enough food and/or significant disruption in water flow/diffusion through the tube. As residence time increases in the tube, so will impaction of the screen.

Figure 9 Detailed feeding protocol for Hormiphora californensis.

Feeds may be skipped when copepods remain in the tube uneaten. When copepods remain in the tube, not consumed, live algae (Isochrysis, Tetraselmis, Rhodomonas etc.) may be added to the next feed as an equal volume to the feed. This helps keep the remaining feed enriched and growing. Feed may be placed in a beaker and slowly dripped into the top of the inner tube using a drip emitter or similar device (DYNALON clamp,part #670715). Algae cultures are drawn from at different stages of growth, so density varies and should be dosed according to appearance of water color in the upper tube. When algae and feed are added slowly they sit on the thermocline and it is quite apparent when existing copepods have eaten the algae and more should be added.

Gelatinous zooplankton require physically stable conditions to grow en masse. Our tube design achieves passive water exchange via diffusion, minimizes fouling of seawater, increases residence time of live prey all with little or no mechanical turbulence. The diffusion cylinder method can be a critical tool for laboratory use in the cultivation of a variety of pelagic gelata species and thus contribute to a range of research in these fragile organisms on population genetics, zooplankton plastivory, behavior and ecology.

Supplemental Information

Supplemental Information 1 Raw data for total length measurements of Hormiphora.

Three columns: the day post hatch the measurement was taken, the length as measured from ImageJ and the flow treatment that measurement came from.

Click here for additional data file.

Supplemental Information 2 Raw water parameters data for T, pH, O2.

The first tab in the file for each flow treatment contains depth profiles using the means and standard deviation. The following tab has all the raw data from 3 trials each.

Click here for additional data file.

We thank Dr. W.E. Browne for his enthusiastic collaboration, comments on draft manuscript and support of ctenophore culturing here at the aquarium. The DEEP-C project for creating a joint ctenophore project that introduced us to Dr. Browne and other ctenophore researchers. Shannon Johnson for her comments on the draft manuscript. John Negrey for instrument calibration and support. John Hoech, Paul Clarkson, Marcus Zevalkink, Sarah Halbrend, Evan Firl, the Drifters gallery staff and volunteers that contributed their time and resources to this project.

Additional Information and Declarations

Competing Interests

Author Contributions

Data Availability

The authors declare that they have no competing interests.

Wyatt L. Patry conceived and designed the experiments, performed the experiments, analyzed the data, prepared figures and/or tables, authored or reviewed drafts of the paper, and approved the final draft.

MacKenzie Bubel performed the experiments, authored or reviewed drafts of the paper, and approved the final draft.

Cypress Hansen conceived and designed the experiments, performed the experiments, analyzed the data, prepared figures and/or tables, authored or reviewed drafts of the paper, and approved the final draft.

Thomas Knowles conceived and designed the experiments, performed the experiments, authored or reviewed drafts of the paper, and approved the final draft.

The following information was supplied regarding data availability:

The raw measurements are available in a Supplemental File.

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
