# Peer review of "Diffusion tubes: a method for the mass culture of ctenophores and other pelagic marine invertebrates"

_PeerJ, doi:10.7717/peerj.8938_

## Round 0.1 · original submission · Major Revisions

The reviewers have commented on your above paper. They indicated that it is not acceptable for publication in its present form.

However, if you feel that you can suitably address the reviewers' comments (included), I invite you to revise and resubmit your manuscript.

·

Basic reporting

The manuscript “Diffusion tubes: a method for the mass culture of ctenophores and other pelagic marine invertebrates” by Patry and collaborators test a method to raise delicate species of pelagic marine invertebrates. I consider the present attempt a good advance in the mass laboratory cultivation of juvenile ctenophores that would facilitate the research with different ctenophore species.

That said, I am adding below few comments and advices.

The authors say that “medium flow outperformed the low flow treatment” (line 215). Can the authors explain in which way? (e.g. length of cydippids, number of cydippids after six weeks). Furthermore, the legend for Figure 6 says that “the high and medium flow treatments ended with significantly greater mean lengths than the low flow treatment” at week 6, while in the text (line196) the authors say that “the medium treatment had the smallest mean lengths”. From the data that the authors have supplied I could see that the mean lengths, after 35 days, were 333 mm and 589 mm for medium and low flow treatments, respectively. Can the authors fix that contradiction?

I encourage the authors to provide more evidence (e.g. growth rate, mortality, fecundity) on the culture of non-ctenophore species (e. g. Nanomia and Oikopleura) with their method. Otherwise, I strongly recommend evaluating if “and other pelagic marine invertebrates” can remain in the title (line3) and abstract (line20) of the article.

The authors supply the number of cydippids that their method can yield as a proof of their method success (line189). Baker and Reeve 1974 (doi.org/10.1007/BF00389086) also provide the fecundity of the animals to test their method to culture ctenophores. Could the authors add some data on animal's fecundity?

The authors include the protocol to culture the ctenophores in the conclusions (lines 250 – 267). I think (IMHO) that a separated box with the protocol will help the reading. Also, a scheme of the culture indicating the initial number of parents, the days that the cydippids spend in the diffusion tube before to be transferred to the Kreisel tank, etc. would help the readers.

Figures and Supplemental material

Figure 4. The legend for Figure 4 should say that the mean of the two measurements taken on opposing sides of the inner tube is a dot, not a line. Besides, the authors should clarify if the line is showing the tendency of the measurements or only is joining the dots.

Figure 6. The title of the Figure 6 is “Pairwise comparison of means for low, medium, and high flow treatments at week 6” and the figure is a box and whisker diagram. As normally box plots depict groups of numerical data through their quartiles and mean is not represented in them, I would like that the authors clarify what is the box and whisker diagram representation, and if the mean is not represented in the diagram, could the authors add it? Otherwise, an easy to confuse reader like myself could get wrong.
Additionally, Could the authors clarify if week 6 is refering 35 days data? If not, Could the authors add the measurements for the week 6 in the supplementary file “Raw data for total length measurements of Hormiphora”?

Supplemental file “Raw water parameters data for T, pH, O2”. In the sheet “High Flow Profile Data” authors say that the turnover rate is 5.5 l/m in the text (line85) is said that is 4.5 Lpm. Can the authors check it?

Other comments.

Line 29. Clarify if citation “Presnell and Browne, 2019” should be “Presnell et al. 2019”.

Line 146. Change “PST” by “PM” or “noon”.

Line 184. Can the authors give the standard error for the probe?

Line 189-190. Change “medium flow (1.1 Lpm)” by “medium flow (2.2 Lpm)” and “low flow (2.2 Lpm)” by “low flow (1.1 Lpm)”.

Line 225. Change “using” by “the use of”

Experimental design

The methods of spawning and culturing section needs more detail. I suggest that the authors improve the description between lines 118 and 147 to include the initial number of cydippids to be raised in each diffusion tube (as animal density can affect the food clearance, and consequently growth rate), how many replicates did they perform for each treatment, how long takes to cydippids to produce eggs (Is there a possibility that juveniles have produced eggs in the diffusion tube during these 6 weeks?). Besides, an estimation of the number of cells of live algae (line 121) (or at least, the optical density of the algae cultures), at the moment of feeding, will help other laboratories to replicate present system. Finally, I also suggest the authors include a short description of how they cultured the other non-ctenophore species (e.g. Nanomia and Oikopleura) at the end of the section (Do you also transfer non-ctenophores animals to Kreisel tanks?). It will be very interesting to see the survival rates of Oikopleura in this system.

I agree with the authors that the initial number of parents might impact in the yield of the culture. The authors say that they obtained higher yields in previous/subsequent trials with more initial parents than the experiment presented in this article (line 222). For that reason, I think that would be worth it to present that data in this article. In the case that these data have been published, they should cite it.

The authors say (line 65) that they have observed several physical parameters including; feeding rates, biofouling, and analyzed fluid flow. However, the results supplied do not include feeding rates or biofouling measurements (that I can see). Have these analyses been done already? If so, make that clearer in the text (results section).

Validity of the findings

The authors should show that they have repeated this experiment more than once (is not a long experiment, 35 days). Specially for the medium flow treatment where they have unexpected results. If they did the experiment more than once, could they make it clear?

Additional comments

In my point of view the reported method seems to be adequate to growth juvenile ctenophores in the lab and could help to spread the use of that phylum as a model organism. However, the conclusions presented in the article are more based in the author’s experience cultivating ctenophores for 4 years (line 246) than in the experiments presented (E.g. the experiment was done with 3 initial parents but the given conclusions are to use 1 to 10 parents).

·

Basic reporting

This manuscript describes the construction of a double cylinder rearing tank, a protocol for raising ctenophores in these tanks, and best practices for mass culture of ctenophores using this new design. Research on ctenophores has been historically limited due to the difficulty of maintaining the animals in the laboratory, but with the availability of genomes and techniques for studying ctenophore biology in the lab, there is a clear need and demand for a solution to culturing these delicate animals. The authors are are world's experts in rearing ctenophores and the insight from this paper will have a substantial impact on researchers and aquarists alike.

* The paper is mostly well organized and well written, but there are some issues regarding the instructions and figures for building the tank. The authors should make building one of these as easy as possible. An IKEA-like step-by-step instruction set should be provided. This should include an illustration of the final product, tools needed, list of parts, and step-by-step assembly procedures. Here is a nice example: https://dokit.io/wp-content/uploads/2019/05/instructions_IKEA_billy.jpg
Figure 1 is an attempt at this, but was very confusing to me as I was trying to build one. It was not step-by-step. For example, the instructions on glueing the mesh (bottom of the inner cylinder was fitted with 55 μm nylon screening) was not detailed enough. In addition to instructions for gluing etc. it should be included how long to wait, importance of soaking materials overnight before usage, etc.

* A table with each part, vendor, part number, approximate cost would be extremely helpful. It would be helpful if parts were lettered and referred to in the diagram and in the "Materials" part of the paper (Table 1A). It would also be helpful to mention where potential replacements could be made and what would be the limits (e.g. 40-75 micron mesh?) either in this table or a supplement.
NOTE: A vendor and part number for the 2 main acrylic tubes should be provided.

* I recommend separating the "Materials" section from an "Assembly" section. The assembly section could walk through the step-by-step figure.

* With so many parts from so many vendors, it is inevitable that part numbers will change and perhaps some parts are no longer carried by particular vendors. To ensure that the manuscript is able to be useful for a long time, the authors could maintain a markdown document on a github repository where the links to particular parts could be kept up to date (and alternatives (perhaps cheaper) from users can be posted). Since GitHub is a software repository, a record of changes is kept and this would enable the historical nature of the paper to be conserved, while allowing the authors to keep an up-to-date set of links, etc.

* building one of these devices costs around $500 USD. It would be useful to mention this up front to save people time who can not afford to make one. Something like: "At the time of the writing of this manuscript, using the specified parts, the cost of building one of these devices is $XXX.YY."

* The model # for the rigid plastic mesh (Model# N1020 80 PentairAES, Inc.) is for a 48" by 50' spool of fencing $202.56. A user needs about .0001% of this and there are far cheaper options available (e.g. https://www.homedepot.com/p/Everbilt-3-4-in-x-3-ft-x-25-ft-Green-PVC-Poultry-Fence-889241EB/205960879 would probably work). Alternatively, a whole could be drilled near the top of the inner cylinder and a rod going through the tube could suspend the inner tube from the rim of the outertube making this mesh unnecessary. In addition the Model #N1670 PentairAES mesh is also for a huge amount and is $96.23 USD. It would be nice if there a smaller quantity/cheaper option. At the very least, it should be mentioned that these will work, but come in large quantities and replacements could be potentially found in local hardware stores?

* Regarding flow rates, it would help to include a table with optimal (or approximate) flow rates per species.

* Line 133 - it would help to have a section heading (e.g. post-diffusion tube rearing) to make it clear that the supplies in this section are not for the diffusion tube. A table 2 with these supplies and a figure showing how they are used would make it more clear.

* Line 140 - it is not clear as written whether Model #24244SC refers to 4mm or 6mm pieces

* Line 191 - These data would be better in a table.

* Figure 5. - Ideally this graph would communicate the difference between the treatments. Something like the boxplot described here: http://t-redactyl.io/blog/2016/04/creating-plots-in-r-using-ggplot2-part-10-boxplots.html
would be more informative.

Line 210: "that is inexpensive" - It would be helpful to contrast the actual price of a kreisel vs. the diffusion tubes.
For example https://aquaticnow.com/schuran/kreisels/breeding-kreisel.html

Line 213: "Using a higher turnover rate..."
-- Should this be "Using the highest turnover rate" since the lower rate was superior to the medium rate?

Line 214: "It is not clear why the medium flow outperformed the low flow treatment"
-- I'm confused. I thought the order was highest flow was best, followed by lowest, then medium?

Line 225: "It is also possible that lower yields may reflect using fewer parents (n = 3) than previous/subsequent trials (n = 10)."
-- Perhaps due to inbreeding depression due to increased levels of self-fertilization?

* A table with recommended inner and outer tube diameters and thickness (with models) for each species would be helpful.
* Likewise, making an updatable csv file of this table in a repo like GitHub that could be updated by the authors as improvements to suggested sizes/flows etc. would be a wonderful addition.

Line 257: "gently remove them from the tube using ladles or dishes"
-- I assume "them" refers to adults, but that is not clear as written.

Experimental design

no comment

Validity of the findings

no comment

---

## Round 0.2 · accepted · Accept

I am pleased to confirm that your paper has been accepted for publication in PeerJ.

Thank you for submitting your work to this journal.

·

Basic reporting

Patry and collaborators have addressed the previous reviewers' comments rightly, improving the clarity and comprehension of their manuscript. However, new typo errata, that authors should correct, have risen during that process:
- The authors say (line 246) that p-value for the Wilcoxon signed-rank test between low and high flows is 0.00046, while in figure 6 is 0.0046.
- Legend figure 7. Change “srages” by “stages”

Experimental design

no comment

Validity of the findings

no comment

Additional comments

no cmment

·

Basic reporting

no comment

Experimental design

no comment

Validity of the findings

no comment

Additional comments

I had 2 very minor points.

1. On line 118 “single diffusion tube setup is ~$465; on line 305 “diffusion tube setup costs ~$300”; Assuming $300 should be $456, should line the phrase on line 306 be changed from “can cost twice that of a diffusion tube setup” be changed to “usually cost more than $600”?
2. Line 315: “Should flows need to be minimized in a future lab setting, perhaps replication will provide better resolution.”
>>> This sentence should be removed. It is redundant with the next sentence and is superfluous as more replicates always provide higher resolution.